# Lightweight Transformer Network for Ship HRRP Target Recognition

**Zhibin Yue, Jianbin Lu * and Lu Wan**

School of Electronic Engineering, Naval University of Engineering, Wuhan 430033, China
* Correspondence: lu_jian_bin@163.com; Tel.: +86-19871160973

**Abstract:** The traditional High-Resolution Range Profile (HRRP) target recognition method has difficulty automatically extracting target deep features, and has low recognition accuracy under low training samples. To solve these problems, a ship recognition method is proposed based on the lightweight Transformer model. The model enhances the representation of key features by embedding Recurrent Neural Networks (RNN) into Transformer's encoder. The Group Linear Transformations (GLTs) are introduced into Transformer to reduce the number of parameters in the model, and stable features are extracted through linear intergroup dimensional transformations. The adaptive gradient clipping algorithm is combined with the Stochastic Gradient Descent (SGD) optimizer to allow the gradient to change dynamically with the training process and to improve the training speed and generalization ability of the model. Experimental results on the simulated dataset show that multi-layer model stacking can effectively extract deep features of targets and raise recognition accuracy. At the same time, the lightweight Transformer model can maintain good recognition performance with low parameters and low training samples.

**Keywords:** Radar Automatic Target Recognition (RATR); High-Resolution Range Profile (HRRP); Recurrent Neural Network (RNN); attention model; lightweight model

## 1. Introduction

HRRP is the vector sum of projections in the target direction after the wideband radar-emitting electromagnetic wave is scattered by the target. It contains target scattering characteristics, geometric structure and attitude, and compared to Synthetic Aperture Radar (SAR) images, HRRP has the advantages of easy acquisition, storage and fast processing [1]. As a result, target recognition using HRRP has gained widespread interest in the field of Radar Automatic Target Recognition (RATR) [2–7].

Discerning how to extract easily distinguishable deep features in HRRP is key to target recognition. Earlier, Du et al. [8] proposed a method to calculate the Euclidean distance in the higher-order spectral feature space to explore the most favourable higher-order spectral features for recognition. Zhang et al. [9] selected the bispectra with maximum interclass differentiability as a feature to avoid the effect of harmful bispectra features on recognition. Molchanov et al. [10] extracted the cepstral coefficients from the micro-Doppler signals of radar echoes and proposed a classification feature based on bicoherence estimation. However, these recognition methods require a priori knowledge of the relevant spectrum and obtaining high recognition rates relies on manual extraction of better classification features.

Since traditional recognition method requires the manual extraction of features and have a limited ability to extract feature representations, the researchers introduced deep learning to automatically extract target features. Wan et al. [11] and Guo et al. [12] applied ordinary Convolutional Neural Network (CNN) and lightweight CNN to the recognition of HRRP, respectively. Subsequently, Chen et al. [13] combined a bi-directional gated recurrent unit (Bi-GRU) with CNN and used Bi-GRU to enhance CNN to extract the valuable features

for recognition, and the experimental results demonstrated that the use of CNN could effectively improve the recognition rate of ship targets.

All the above models ignore the fact that HRRP signals are temporal signals, and in order to make better use of the temporal dependence in HRRP, Liu et al. [14] applied recurrent neural networks (RNN) to HRRP recognition and achieved better results under small sample conditions. However, the RNN was also found to suffer from the problem of gradient vanishing. To solve this problem, Zhang et al. [15] proposed a VGM-RNN model to extrapolate missing samples in the sequence, which not only effectively alleviates the problem of gradient vanishing, but also has excellent noise immunity. The literature [16–18] proposes to combine the attention mechanism with RNN, Long Short-Term Memory (LSTM) and Gated Recurrent Units (GRU). According to the importance of the implicit state features in target recognition, weights are assigned to the implicit state features encoded by RNN, LSTM and GRU to make the extracted features more discriminative. Du et al. [19] combined RNN with a deep clustering mechanism, using region decomposition and attention mechanism to accumulate each region weight, while optimizing the cross-entropy loss function to make the model more robust and recognition performance better. The literature [20] used attention modules and feedforward neural units to construct Transformer networks, and extract deep features by stacking attention modules and feedforward neural units [21,22].

This paper improves the encoding part of the Transformer model and proposes a lightweight Transformer model. The model is based on the Transformer encoders and introduces a local RNN module [23] at the input of each encoder to extract correlations between neighbouring distance units, and combines the global correlations extracted by the Transformer encoders to enhance the representation of feature. At the same time, intergroup dimensional transformations and feature interactions by GLTs [24] are used to fully extract the target features and reduce parameters in the model. The use of adaptive gradient clipping algorithm [25] in the SGD optimizer allows the gradient descent process to vary dynamically with model training, speeding up model training process and enhancing the stability of model.

## 2. Proposed Model

### 2.1. Lightweight Transformer Model

The encoder in the Transformer model is improved in this paper, and the structure of the three-layer lightweight Transformer model is shown in Figure 1. The lightweight Transformer model uses the pre-processed HRRP adjacent to k time steps as input through the local RNN module to extract target local temporal feature, and uses residual connection to alleviate the gradient vanishing problem during RNN training process. Combining GLTs with Transformer encoder: the number of model parameters is reduced by transforming the dimension of the input side of the self-attention unit and the feedforward neural network unit, and the features are fully extracted in different dimensions. During model optimization process, the SGD optimizer is combined with adaptive gradient clipping algorithm, so that the results of the training process can be fed back to the gradient descent process to speed up the training speed and enhance model stability.

### 2.2. Data Pre-Processing

The raw HRRP data of radar targets have aspect sensitivity, amplitude sensitivity, and time-shift sensitivity [26], which have a significant impact on model recognition performance. Therefore, before using the model to recognize, this paper applies Euclidean norm to the raw ship target HRRP to remove the amplitude sensitivity. Suppose the HRRP data after amplitude normalization is:

$$\boldsymbol{p} = [p_1, p_2, \ldots, p_L]^T \tag{1}$$

where $L$ is the number of distance units and $p_i$ is the normalized amplitude of the $i$th distance unit. When using HRRP data for target recognition, the time domain HRRP is converted into sequence data and then input into the model for target recognition.

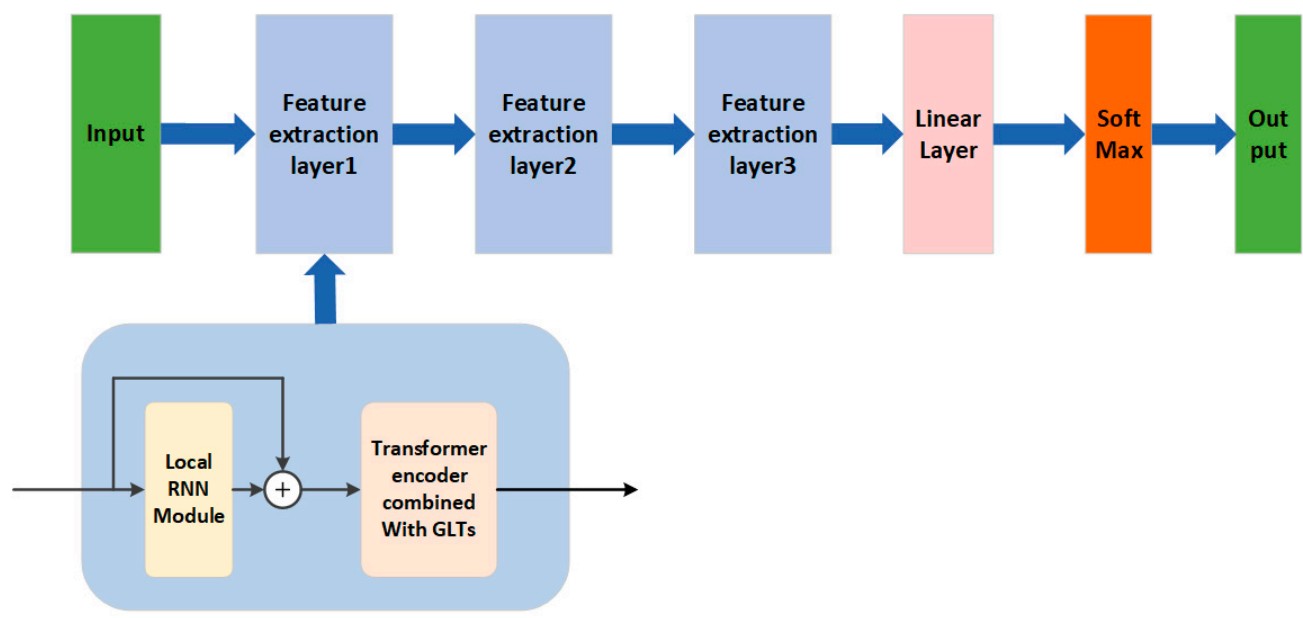

**Figure 1.** Lightweight Transformer model structure. The feature extraction module consists of a local RNN module and a Transformer encoder modified by GLTs. The classifier consists of a linear layer and a SoftMax function.

We use the sliding window method proposed in the literature [27] to transform the HRRP data into serial data after amplitude normalization in the following steps. The amplitude normalized HRRP data is intercepted as a short sequence containing multiple range cells by means of sliding window.

$$X = [x_1, x_2, \ldots, x_T] \tag{2}$$

$$x_l = \left[ p_{(l-1)c+1}, p_{(l-1)c+2}, \cdots, p_{(l-1)c+d} \right] \tag{3}$$

where $x_l \in \mathbb{R}^{d \times 1}$ is the input at moment $l$ with length $d$. The matrix $X \in \mathbb{R}^{d \times T}$ corresponds to $T$ time steps with data of dimension $d$ at each time step.

### 2.3. Local RNN Module

The local RNN module aims at extracting local features and makes full use of the advantages of RNN in short sequence feature extraction. With a short sequence consisting of three time steps as a single RNN input, the structure of the local RNN module is shown in Figure 2. The white circles represent the zero sequence, the blue circles represent the original input sequence, and the light blue circles represent the output sequence. The white circles and blue circles form the input of the local RNN module.

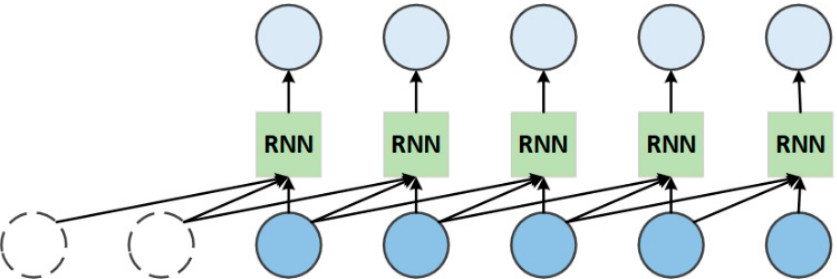

**Figure 2.** Local RNN module.

Take the sequence containing $k$ time steps in HRRP as the input of a single RNN as an example. The pre-processed HRRP sequence containing $T$ time steps is $X = [x_1, x_2, \ldots, x_T]$. A traditional RNN takes a sequence of $T$ time steps and inputs them in time order to obtain a hidden state vector characterizing the target feature. Since RNN has the problem of gradient vanishing when processing long sequences, the local RNN module is used to alleviate the gradient vanishing problem while extracting local features from short sequences. The local RNN module is essentially extracting correlation features between neighbouring distance units within the HRRP sequences. The local RNN module first complements the left side of the input sequence $X$ with $k-1$ zero vectors according to the set short sequences size $k$ to form a new input sequence $X'$.

$$X' = \left[ \underbrace{0, \ldots, 0}_{k-1}, x_1, x_2, \ldots, x_T \right] \tag{4}$$

The sliding window of size $k$, sliding through the distance of 1 from left to right in turn, forms $T$ short sequences, each containing $k$ time steps in $X'$. The $T$ short sequences are fed into $T$ RNNs, each of which outputs a hidden state vector characterizing the feature, and the $T$ hidden state vectors are stitched into a new hidden state vector.

$$h_t = RNN(x_{t-k+1}, x_{t-k+2}, \ldots, x_t) \tag{5}$$

$$H = [h_1, h_2, \ldots, h_T] \tag{6}$$

Adding a local RNN module before Transformer's encoder can make up for Transformer's disadvantage in local feature extraction, and act as a positional encoding. The features extracted by the improved model contain more information and are more comprehensive and more generalizable.

*2.4. Group Linear Transformations (GLTs)*

For a model, a larger number of parameters means higher equipment requirements and longer training time. A comparison of the Transformer encoder and the Transformer encoder combined with GLTs is shown in Figure 3. Compared with the standard Transformer encoder structure, the Transformer encoder combined GLTs uses GLTs at the input to process the input data. In two stages of expansion and reduction, the input data are first mapped to a high-dimensional spatial expansion to obtain more easily distinguishable feature, and then the dimension is reduced as the input to the Self-Attention layer. This transformation allows for more efficient parameter assignment in the expansion and reduction stages, yielding better results than simple linear layer coding. This is because GLTs are able to learn a wider range of features in different dimensions. Under the condition of ensuring good recognition accuracy, Multi-head Attention can be replaced by Self-Attention, while the input dimension of Self-Attention is half of that of Multi-head Attention in the standard Transformer. So, the number of multiplication and addition operations required for calculation is reduced by half, and the corresponding computational effort is greatly

reduced. The input dimension of the lightweight feedforward neural network in the Transformer encoder combined with GLTs is the same as that of the standard Transformer, but the dimension reduction process in the first layer reduces both the dimension and the size of the feedforward neural network to one-fourth of the original size, and the parameters become one-sixteenth of the original size.

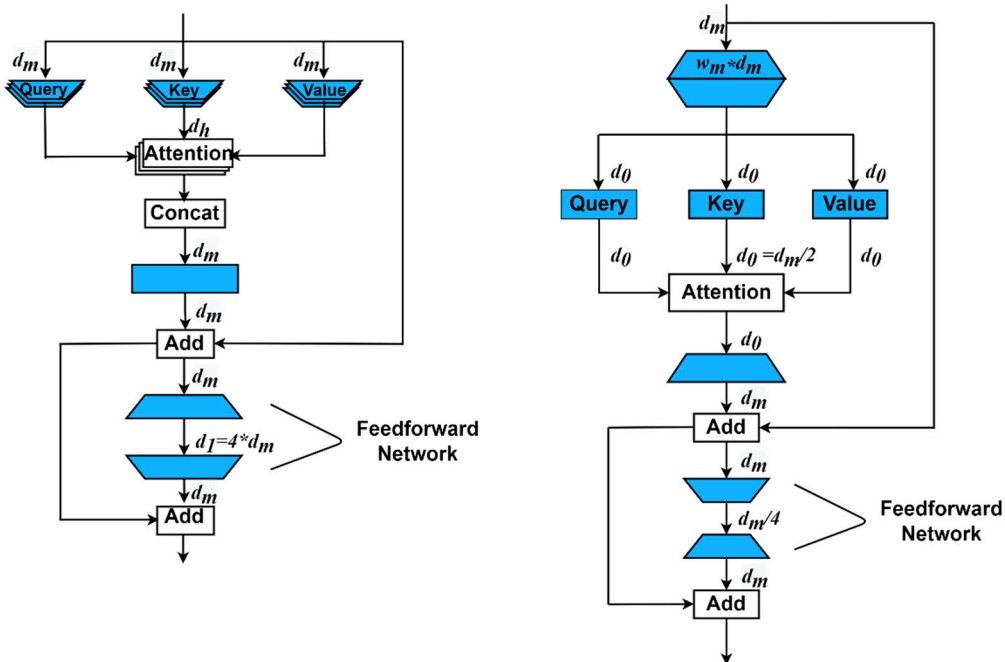

**Figure 3.** Comparison of Transformer encoder and Transformer encoder combined with GLTs. The left is the standard Transformer encoder. The right is the Transformer encoder combined with GLTs.

The hidden vector $H$, which is output by the local RNN module, is used as the input to the Transformer encoder with the GLTs. Suppose the number of layers of GLTs is $N$. In the expansion stage, in $\lceil N/2 \rceil$ linear layers, the $d_m$-dimensional input is projected into the high-dimensional space with $d_{\max} = w_m d_m$, where $w_m$ is expanded dimensional coefficients; similarly, in the reduction stage, the $d_{\max}$-dimensional vector is projected into the $d_0$-dimensional space using the remaining $N - \lceil N/2 \rceil$ layers, projecting the $d_{\max}$ dimensional vector into the $d_0$ dimensional space.

$$g^l = \begin{cases} \min\left(2^{l-1}, g_{\max}\right), 1 \le l \le \lceil N/2 \rceil \\ g^{N-1}, Otherwise \end{cases} \tag{7}$$

where $g_{\max}$ is the set maximum number of linear transformation layers per layer. $g^l$ denotes the number of linear transformation layers contained in the $l$th layer.

Input the hidden vector $H$:

$$Y^l = \begin{cases} F\left(H, W^l, b^l, g^l\right), l = 1 \\ F\left(MIX\left(H, Y^{l-1}\right), W^l, b^l, g^l\right), Otherwise \end{cases} \tag{8}$$

where $W^l = \left[W_1^l, W_2^l, \ldots, W_{g^l}^l\right]$ and $b^l = \left[b_1^l, b_2^l, \ldots, b_{g^l}^l\right]$ are the weights and bias of the $g^l$ group linear transformation function in the $l$th layer, respectively. $F$ is a linear transformation function that divides the input $H$ into $g^l$ non-overlapping groups so that $H$ can be written as $H = \left[H_1, H_2, \ldots, H_{g^l}\right]$, and the linear layer transforms each $H_i$ by the weights $W_i^l$ and $b_i^l$ to obtain an output $Y_i^l = H_i W_i^l + b_i^l$. The output $Y_i^l$ of each group is then stitched to produce the final output $Y^l$.

GLTs combine the output $Y^{l-1}$ with the input $H$ through the mixer *MIX* to mix the features and extract more stable features of the target, which can also avoid the problem of gradient vanishing.

$$MIX\left(H, Y^{l-1}\right) = shuffle\left[H, Y^{l-1}\right] \tag{9}$$

Due to the superior performance of GLTs, Self-Attention is used instead of Multi-head Attention to obtain the same attention effect and achieve a better recognition accuracy using smaller dimensions and fewer operations to calculate the attention scores and carry out the assignment of weights:

$$\boldsymbol{u} = self\_Attention\left(\boldsymbol{Y^l}\right) = Attention(\boldsymbol{Q}, \boldsymbol{K}, \boldsymbol{V}) = softmax\left(\frac{\boldsymbol{QK^T}}{\sqrt{d_k}}\right)\boldsymbol{V} \tag{10}$$

where $\boldsymbol{Q}$, $\boldsymbol{K}$, and $\boldsymbol{V}$ are three key-value vectors, which are abbreviations of *Query*, *Key*, and *Value* vectors, respectively. $d_k$ is the dimension of the $\boldsymbol{K}$ vector, and $\boldsymbol{Q}$ and $\boldsymbol{K}$ are obtained by linear transformation of the output of the previous layer, and the final attention values are obtained by calculating the attention distribution between $\boldsymbol{Q}$ and $\boldsymbol{K}$ and attaching it to $\boldsymbol{V}$.

The feedforward neural network layer contains two linear layers and a nonlinear activation function for feature transformation of the attention scores, where the nonlinear activation function is using GELU and the dimensionality of the linear layer is changed by first downscaling to one-fourth of the input dimension and then upscaling to the original dimension:

$$\boldsymbol{v} = Feedforward(\boldsymbol{u}) = ((\boldsymbol{uW_1} + \boldsymbol{b_1}) \times \phi(\boldsymbol{uW_1} + \boldsymbol{b_1}))\boldsymbol{W_2} + \boldsymbol{b_2} \tag{11}$$

where $\phi(x)$ is the cumulative distribution function of the normal distribution, $\boldsymbol{W_1}$, $\boldsymbol{b_1}$ are the weights and biases of the linear layer of the reduced-dimensional part, and $\boldsymbol{W_2}$, $\boldsymbol{b_2}$ are the weights and biases of the linear layer of the raised-dimensional part.

Supposing that the Transformer encoder module with GLTs has $M$ layers, the output $\boldsymbol{v^M}$ obtained after the last layer is decoded by a network containing a single linear layer, and finally normalized by the SoftMax function to obtain the final output.

$$\boldsymbol{y} = Linear\left(\boldsymbol{v^M}\right) \tag{12}$$

$$\boldsymbol{y'} = softmax(\boldsymbol{y}) \tag{13}$$

where $\boldsymbol{y'}$ denotes the probability vector that the sample belongs to each class of ship targets.

*2.5. Adaptive Gradient Clipping*

In this paper, an adaptive gradient clipping algorithm is used in the process of optimization. Prem Seetharaman et al. [25] proposed a simple method to adaptively select the gradient clipping threshold based on the numerical history of the gradient descent process observed during the training process, resulting in smoother loss function curves and more robust recognition performance.

Suppose the objective function is $f(\boldsymbol{X}; \theta)$, where $\boldsymbol{X}$ is the input data and $\theta$ is a parameter, and given a learning rate $\lambda$, the parameter $\theta$ is updated iteratively from $t-1$ to $t$ with gradient descent as defined below.

$$\theta_t = \theta_{t-1} - \lambda h_c \nabla_\theta f(\boldsymbol{X}, \theta_{t-1}) \tag{14}$$

$$h_c = \min\left\{\frac{\eta_c}{\|\nabla_\theta f(X, \theta_{t-1})\|}, 1\right\} \tag{15}$$

where $\eta_c$ is a clipping hyperparameter that is set according to the model. In clip by norm, the entire gradient is scaled if the norm of the gradient exceeds a threshold.

The function of the adaptive gradient clipping algorithm is to let $\eta_c$ follow the model training process dynamically by setting a new hyperparameter $p$, representing percentile cut-off for setting $\eta_c(t)$ to cut the gradient parametrization. In a training process, $X_t \in D$, $D$ represents the training dataset and $t$ is one of the training batches. Supposing that $G_h$ stores the gradient norm of each batch and the initial $G_h(0) = [\ ]$.

$$g_h(t) = \nabla_\theta f(X_t; \theta_{t-1}) \tag{16}$$

The gradient norm is added to $G_h$ for each batch computed, i.e., during the training of the $t$th batch, there are:

$$G_h(t) = [g_h(0), g_h(1), \ldots, g_h(t-1), g_h(t)] \tag{17}$$

Calculate the gradient norm of the $p$th percentile value in the current gradient vector.

$$\eta_c(t) = p \times G_h(t) \tag{18}$$

The gradient cropped $\theta_t$ is obtained by bringing $\eta_c(t)$ into (15).

## 3. Experimental Results and Analysis

### 3.1. Introduction to the Dataset

Most of the ship targets are non-cooperative targets, and it is difficult to build an HRRP database of the targets from the actual measurement data. In this paper, we use CAD3D software to build ten kinds of 1:1 ship target models and import them into CST electromagnetic simulation software. The structural parameters of ten kinds of ship targets are shown in Table 1. CST simulation parameters are set as follows: azimuth angle is 0°–360°, pitch angle is 90°, angle step is 1°; radar centre frequency is 3 GHz, bandwidth is 150 MHz, polarization mode includes vertical polarization and horizontal polarization, frequency sampling points are 360, the default optimal grid profiling size of the software is used, and ray tracing algorithm is selected for the solution. Eventually, the HRRP data of 360 azimuths for ten types of ship targets are simulated.

**Table 1.** Structural parameters of ten types of ship targets.

| Ship Number | Length (m) | Width (m) |
|:---:|:---:|:---:|
| 1 | 182.8 | 24.1 |
| 2 | 153.8 | 20.4 |
| 3 | 162.9 | 21.4 |
| 4 | 99.6 | 15.2 |
| 5 | 332.8 | 76.4 |
| 6 | 337.2 | 77.2 |
| 7 | 17.1 | 3.6 |
| 8 | 143.4 | 15.2 |
| 9 | 17.6 | 4.5 |
| 10 | 16.3 | 4.7 |

The lack of training samples can lead to overfitting of the model, and the number of HRRP simulated with CST is far from sufficient for the proposed model, which needs to be expanded. In this paper, the original data are expanded by adding the clutter conforming to the K distribution in three times by the size of the signal-to-clutter ratio of 10 dB, and the data are expanded by a factor of three at each signal to clutter ratio. The HRRP of the ten-class ship target under the 10 dB signal-to-clutter ratio condition is shown in Figure 4. The HRRP of a class ten ship target under 10 dB signal-to-clutter ratio for both polarizations is shown in Figure 4.

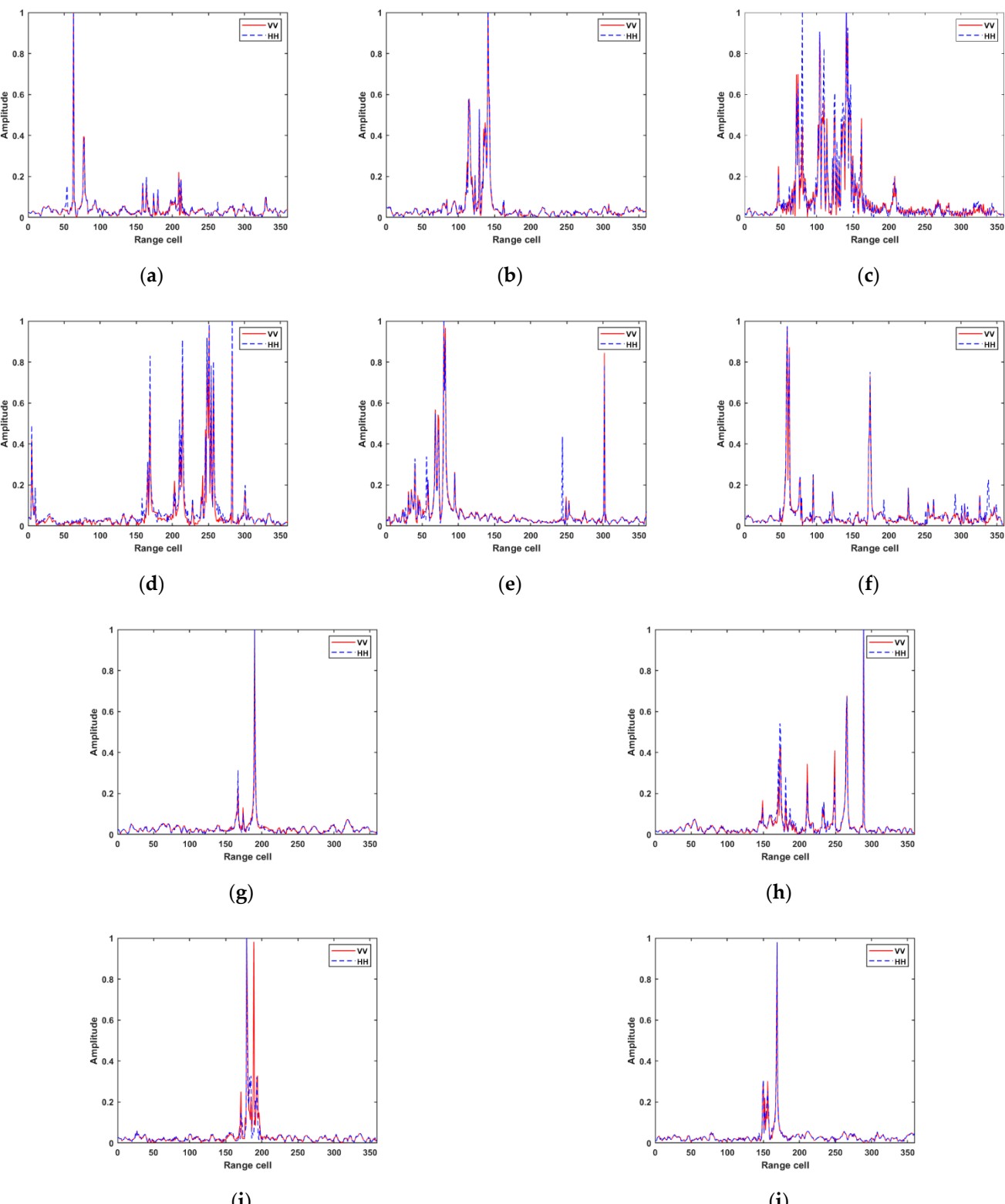

**Figure 4.** HRRP of class ten ship targets. (**a**–**j**) represent ships one–ten.

The model parameters are set: the total number of training epochs is 300, the initial learning rate is 0.03, and the learning rate becomes one-fourth of the original one for every 80 iterations during the training process. The dimensions of the hidden layers are all set to 128. The optimizer is chosen to be Stochastic Gradient Descent (SGD) with a batch size of 32. The sliding window length of the preprocessing is $d = 48$ and the sliding distance

$l$ = 24. The kernel function used for Support Vector Machines (SVM) is the Radial Basis Function. The role of Principal component analysis (PCA) is to reduce the input data to two dimensions.

### 3.2. Recognition Performance

To explore the effect of the number of stacked layers on recognition performance, the number of layers one to eight are taken, and the relationship between recognition accuracy and the number of layers is shown in Figure 5. For recording purposes, we denote the local RNN module as LR and the adaptive gradient clipping algorithm as AC.

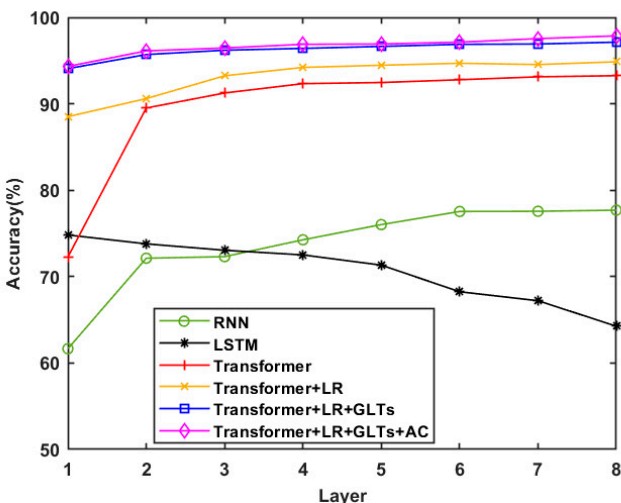

**Figure 5.** Relationship between the number of layers and the recognition accuracy.

As can be seen in Figure 5, except for the LSTM model, the rest of the models are able to learn deeper features as the number of stacked layers increases, and the recognition performance of the models is improved. The Transformer model with the addition of the local RNN module extracts the correlation between neighbouring distance units of HRRP, which makes the feature representation more powerful and the model can achieve more than 88% accuracy even at a lower number of layers. With the addition of GLTs, more stable features are extracted by projecting the features in different dimensions. The model achieves more than 94% recognition accuracy at lower layers, which is much better than other models, and the model is no longer sensitive to layer changes. Compared with other models, the accuracy variation curve with the number of layers is smoother. To verify the best performance of the model, all models below use eight layers.

Table 2 lists the recognition accuracy of various models. As can be seen from the results in Table 2, the recognition accuracy achieved by the machine learning method SVM and the SVM method after PCA dimensionality reduction is not very satisfactory. In contrast, the recognition accuracy is higher by the method of building neural networks. Compared with the traditional RNN and LSTM models, which are not ideal for the recognition of ship target HRRP, the recognition accuracy of the Transformer model is greatly improved. Compared with traditional Recurrent Neural Networks, the attention module in Transformer can better learn the important features in the sequence data, allowing the model to adaptively search for feature regions important for recognition and amplify the significant feature regions that can distinguish different targets, instead of aimlessly matching all regions of targets. With the addition of the local RNN module, RNN is used to focus the correlation between neighbouring distance units to extract local features, which are complementary to Transformer for overall feature extraction, and the recognition accuracy is raised. With the addition of GLTs, the features extracted by the local RNN module are fed into the linear transformation of the N-layer to realize the mapping of the input vector from low-dimensional to high-dimensional, allowing more features to be represented. Then, the

high-dimensional easily distinguishable feature is reduced to the input dimension of Self-Attention by linear transformation of N layers. At the same time, feature exchange between groups is performed in the linear transform group to achieve global feature learning, which enhances the data expression and effectively raises the recognition accuracy. The adaptive gradient clipping algorithm is used to optimize the gradient descent process so that the gradient can obtain the optimal solution within a reasonable range, and the recognition accuracy is raised by a small margin. Compared with Transformer, the recognition accuracy is raised by 1.61% with the addition of local RNN module, by 3.88% with the addition of LR and GLTs, and by 4.61% with the addition of LR, GLTs and AC.

**Table 2.** Recognition performance of six different models.

| Models | Accuracy (%) |
|---|---|
| SVM | 46.52 |
| PCA + SVM | 49.21 |
| RNN | 77.69 |
| LSTM | 64.26 |
| Transformer | 93.25 |
| Transformer + LR | 94.86 |
| Transformer + LR + GLTs | 97.13 |
| Transformer + LR + GLTs + AC | 97.86 |

Figure 6 shows the curves of the relationship between the model recognition accuracy and epoch. As the number of epochs increases, the recognition accuracy of each model subsequently raises. Compared with traditional RNN and LSTM models, Transformer has faster convergence and less fluctuation after the model reaches stability. Adding the local RNN module, the ability of the model to extract more expressive features is highlighted after 50 epochs, and the recognition accuracy is raised. On this basis, the advantage of extracting more easily distinguishable features is obvious with the addition of GLTs. With the addition of adaptive gradient clipping algorithm, the recognition accuracy raised significantly in the first few epochs, the convergence of the model was significantly faster, and the model had more stable performance.

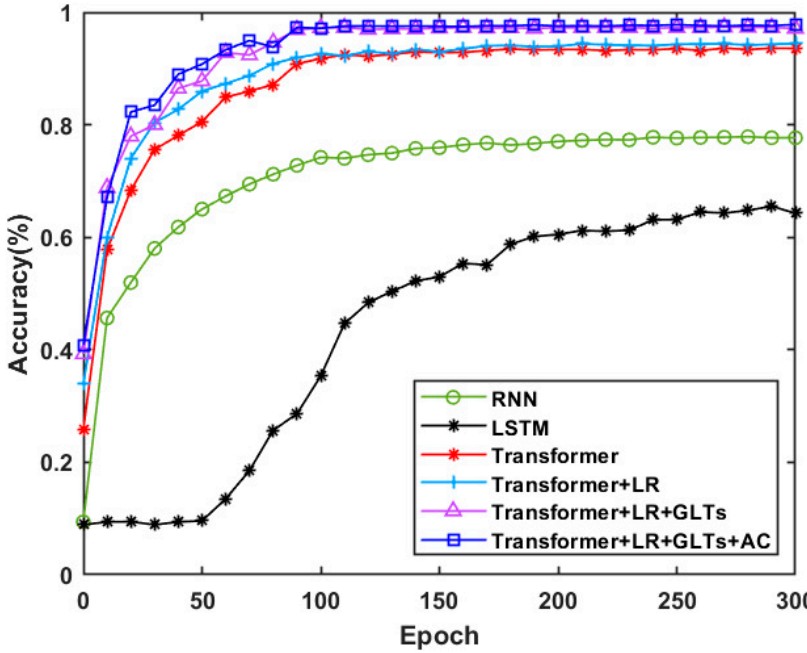

**Figure 6.** Relationship between recognition accuracy and epoch.

### 3.3. Feature Visualization

To verify the separability of the features extracted by different models, the original HRRP data and the features extracted by the six different models were input to the two-dimensional ISOMAP for visualization, and the results of feature visualization are shown in Figure 7. The data points in ten colours in Figure 7 represent ten types of targets. In Figure 7a, the ten types of ship targets in the original HRRP data are basically overlapped together, and the separability is poor. In Figure 7b,c, the feature separability extracted by RNN and LSTM models is better than Figure 7a, and the spacing between target samples of different categories increases, but still, most of the features overlap and are still inseparable. Figure 7d,e shows significantly better feature extraction after Transformer and Transformer + LR models than Figure 7b,c, with better aggregation of the same category targets and increased spacing between samples of different category targets, making the features of each target separable. However, the distances of samples within the same category are greater than the distances of samples between different categories, resulting in fuzzy boundaries of different target features. In Figure 7f,g, the features extracted by Transformer + LR + GLTs and Transformer + LR + GLTs + AC models are separable, with clearer boundaries between classes and almost no overlap.

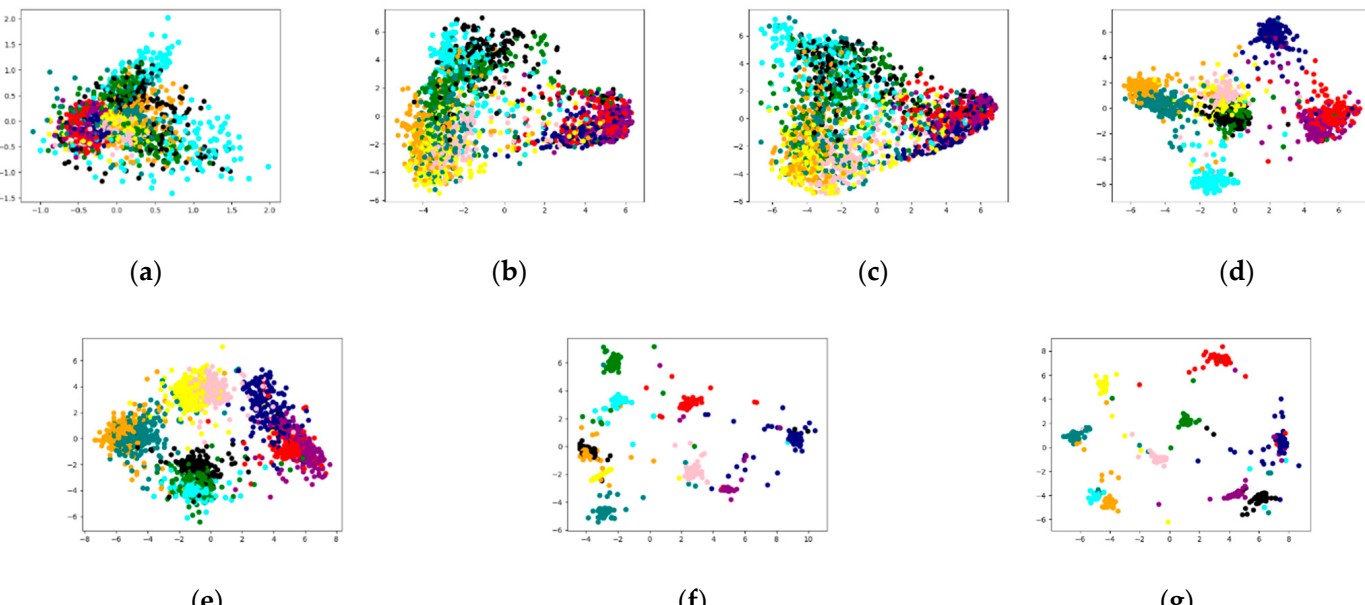

**Figure 7.** Two-dimensional ISOMAP projection. a-h are the two-dimensional visualization results of the extracted features of several models. (**a**) original data, (**b**) RNN, (**c**) LSTM, (**d**) Transformer, (**e**) R-Transformer, (**f**) R-DeLighT, (**g**) R-DeLighT + Autoclip.

### 3.4. Effect of Training Set Size

A robust recognition model should be able to maintain good recognition performance even with a small number of training samples. To test the effect of the size of the training sample on the recognition rate, six size training sets were used in the experiments, namely 4320, 8640, 10,800, 12,960, 17,280 and 19,440, corresponding to 20%, 40%, 50%, 60%, 80% and 90% of the data set, respectively. The results of the six types of models on different training sets are shown in Table 3.

**Table 3.** Accuracy with different training set sizes (%).

| The Proportion of Training Set | 20% | 40% | 50% | 60% | 80% | 90% |
|:---:|:---:|:---:|:---:|:---:|:---:|:---:|
| RNN | 50.05 | 64.94 | 68.17 | 70.25 | 75.57 | 77.69 |
| LSTM | 31.29 | 43.79 | 49.32 | 54.29 | 61.34 | 64.26 |
| Transformer | 65.72 | 78.71 | 83.12 | 85.33 | 91.71 | 93.25 |
| Transformer + LR | 69.03 | 82.43 | 85.86 | 88.40 | 92.69 | 94.86 |
| Transformer + LR + GLTs | 81.83 | 87.46 | 90.10 | 92.60 | 95.67 | 97.13 |
| Transformer + LR + GLTs + AC | 82.53 | 87.97 | 90.88 | 93.23 | 96.13 | 97.86 |

As can be seen from the table, with only 20% of the training samples, the local RNN module, GLTs combined with the Transformer model can achieve 81.83% accuracy, which is a 16% raise in accuracy compared to the Transformer. In the process of changing the training set from 90% to 20%, the accuracy of Transformer model decreased by 27.53%, while the local RNN module, GLTs combined with Transformer only decreased by 15.3%, indicating that the model is more stable in the process of changing the training set. Transformer combined with the local RNN module can raise the recognition accuracy by 1–3% when the training set is reduced compared to Transformer model. Therefore, in comparison, the GLTs can make the Transformer model more robust when the training set is reduced.

### 3.5. Model Complexity Analysis

The complexity of the model is intuitively reflected by the number of parameters and the amount of computation. The number of parameters and the amount of computation for six different model are shown in Table 4.

**Table 4.** Model parameters and flops.

| Model | Params (K) | Flops (G) | Accuracy (%) |
|:---:|:---:|:---:|:---:|
| RNN | 255.27 | <0.01 | 77.69 |
| LSTM | 954.23 | <0.01 | 64.26 |
| Transformer | 1130.21 | 0.01 | 93.25 |
| Transformer + LR | 1531.67 | 0.03 | 94.86 |
| Transformer + LR + GLTs | 292.94 | <0.01 | 97.13 |
| Transformer + LR + GLTs + AC | 292.94 | <0.01 | 97.86 |

From Table 4, it can be seen that the Transformer model combined with the local RNN module and GLTs, is about the same as the RNN model in terms of parameters, compared to the Transformer model, which has only a quarter of parameters, and moreover, only a fifth of parameters of the Transformer model combined with the local RNN module. In terms of computation, the Transformer model combining local RNN modules and GLTs is about the same as the RNN and LSTM models, much smaller than the Transformer and the Transformer model combined with local RNN module, achieving the best recognition rate using fewer parameters and computation. Model complexity is proportional to the number of parameters, so combined with the local RNN module and GLTs, the lightweight Transformer model achieves a win-win situation between complexity and accuracy.

## 4. Conclusions

In this paper, a lightweight Transformer network is proposed for recognizing ship targets. Through experimental studies, the model is able to enhance the effect of feature extraction and improve the recognition accuracy of the model with a smaller number of parameters. Under the same experimental conditions, the number of parameters and computational effort are greatly reduced and the accuracy is improved by at least 3.8 percent compared to other models. In future research, we will extend the application of the model to speech recognition, heart sound classification, picture recognition and other field, and compare it with other classical deep learning models. At the same time, we will continue

to improve the model, adding functions such as rejection of targets outside the database, alarming of abnormal targets, etc., to expand the application scope of the model.

**Author Contributions:** Conceptualization, Z.Y. and J.L.; methodology, Z.Y.; software, Z.Y.; validation, L.W. and J.L.; formal analysis, Z.Y.; investigation, Z.Y.; resources, J.L.; data curation, L.W.; writing—original draft preparation, Z.Y.; writing—review and editing, J.L.; visualization, L.W.; supervision, J.L. All authors have read and agreed to the published version of the manuscript.

**Funding:** This research was funded by the National Natural Science Foundation of China, grant number 61501486.

**Institutional Review Board Statement:** Not applicable.

**Informed Consent Statement:** Informed consent was obtained from all subjects involved in the study.

**Data Availability Statement:** Data are available upon reasonable request to the submitting author.

**Conflicts of Interest:** The authors declare no conflict of interest.

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
