# Peer review of "Lightweight Transformer Network for Ship HRRP Target Recognition"

_applsci, doi:10.3390/app12199728_

Round 1
Reviewer 1 Report
Various simulated HRRP results of the ship used for target recognition needs to be added, including different polarization and azimuthal angle dependences.
Figure 4 needs to be made correction to show HRRP characteristics according to different V-polarization & H-polarizations.
In the conventional ATR, the mean & standard deviations of the interval between adjacent peaks and the adjacent spacing in HRRP are already used as feature vectors.
It is necessary to add a comparison of recognition performance in case that the machine learning based these feature vectors such as PCA and SVM and your proposed Transformer + local RNN model.
P.9, line 285 sentences are broken. The sentence needs to be correct.
Author Response
Dear Editor and Reviewers:
On behalf of my co-authors, we are very grateful to you for giving us an opportunity to revise our manuscript. We are appreciate you very much for your positive and constructive comments and suggestions on our manuscript.
Please refer to the attachment for specific responses and revisions I have made in response to your questions.
Best regards.
Zhibin Yue

Reviewer 2 Report
Dear Authors,
You did very good job working on this research topic. It is really important to present algorithms reducing time and complexity of the computations. Presented method seems to be good for practical use in the autonomic ship navigation systems. I suggest trying to apply it in a practical research as a next step of Your work.
Your workflow and results are presented in a detailed and an orderly manner. But I suggest rewriting Conclusions, becouse they sound rather than Abstract of the paper. In my opinion they should be supported by the results and expanded.
Moreover I have some remarks:
1. Sentence in lines 25-26 is difficult to understand. I suggest dividing it into two separate ones.
2. What does abbreviation SAR stand from in lin 27?
3. Line 83 Transformer... transform... It is obvious that transformer transforms, I suggest changing word.
4. Figure 1. - There is a lack of signal names (unmarked arrows). Three blue boxes in the second line are unnamed. I suggest marking classifier described as a connection of two boxes in the text.
5. Line 104 "method of literature" - is not clear formulation for me.
6. Lines 106-107 "multiple distance units by the sliding window" is also unclear formulation for me. I suggest describing it in a different way.
7. Figure 2. - I suggest for better readibility point out what does each symbol mean (light blue, blue and white circles).
8. Line 170 - there are big spaces between N-[N/2].
9. Figure 3. Names "Querry", "Key", "Value" and "Attention" are hard to read, due to too big font for the frames. Annotations d1 and d0 are too close to arrows.
10. In line 174 there is a lack of dot.
11. In line 215 there is a lack of citation, only nae is presented, without the citation number.
12. In Fig.5 and Fig.6 red lines' colours are hard to distinguish. I suggest changing their colurs or line types.
13. In Fig.7 descriptions (a, c, d, e, g, h) there are no spaces after ")'. Moreover there is a lack of Fig.7(f).
Author Response

(The authors gave the same response as above.)

Round 2
Reviewer 2 Report
Thank You for Your revision. Now presented article is really good quality one.